# TCR microclusters form spatially segregated domains and sequentially assemble in calcium-dependent kinetic steps

Jason Yi[1], Lakshmi Balagopalan[1], Tiffany Nguyen[1], Katherine M. McIntire[1] & Lawrence E. Samelson[1]

Engagement of the T cell receptor (TCR) by stimulatory ligand results in the rapid formation of microclusters at sites of T cell activation. Whereas microclusters have been studied extensively using confocal microscopy, the spatial and kinetic relationships of their signaling components have not been well characterized due to limits in image resolution and acquisition speed. Here we show, using TIRF-SIM to examine the organization of microclusters at sub-diffraction resolution, the presence of two spatially distinct domains composed of ZAP70-bound TCR and LAT-associated signaling complex. Kinetic analysis of microcluster assembly reveal surprising delays between the stepwise recruitment of ZAP70 and signaling proteins to the TCR, as well as distinct patterns in their disassociation. These delays are regulated by intracellular calcium flux downstream of T cell activation. Our results reveal novel insights into the spatial and kinetic regulation of TCR microcluster formation and T cell activation.

[1] Laboratory of Cellular and Molecular Biology, Center for Cancer Research, National Institutes of Health, Bethesda, MD 20892, USA. Correspondence and requests for materials should be addressed to L.B. (email: balagopl@mail.nih.gov) or to L.E.S. (email: samelsonl@mail.nih.gov)

T cell activation is mediated by engagement of the TCR, which consists of the α and β chains, the CD3δ, ε, γ, and TCRζ subunits. Recognition and binding of peptide-loaded major histocompatibility complex protein (pMHC) by the α and β TCR chains initiates the signal transduction cascade by recruiting Src-family protein tyrosine kinases (PTKs), predominantly Lck, or Fyn, to phosphorylate the immunoreceptor-based tyrosine activation motifs (ITAMs) on the intracellular CD3 and TCRζ subunits of the TCR. Phosphorylation of the ITAMS leads to the binding and activation of a Syk-family PTK, zeta-chain-associated protein kinase 70 (ZAP-70), which in turn phosphorylates key adapter proteins, including the transmembrane protein, linker of activation of T cells, or LAT[1,2]. LAT contains several tyrosines, which, after phosphorylation, can bind Src homology (SH2)-containing molecules, notably GADS, GRB2, and PLCγ1. This LAT complex subsequently recruits other adapters and enzymes, including SLP76, VAV1, NCK, and ADAP. Thus, LAT serves as an important scaffold for the recruitment of multiple downstream effectors involved in TCR signal transduction.

T cells display remarkable sensitivity to antigen despite the relatively weak affinity of TCRs for pMHCs and low numbers of stimulatory ligand on the antigen presenting cell (APC) surface[3,4]. This sensitivity is thought to be, in part, the result of signal amplification from the transiently engaged TCRs through a multi-protein structure at the membrane called the TCR microcluster[5]. Within seconds of T cell engagement to an activating surface, submicron-sized clusters marked by the TCR and other signaling molecules form at the contact site and act as a platform for the recruitment and activation of downstream effector molecules[6]. Studies using anti-TCR-coated coverslips or pMHC-containing lipid bilayer to activate T cells have shown concentrated tyrosine phosphorylation activity, as well as dynamic localization of TCRζ, ZAP70, and LAT to these microclusters, indicating that the TCR microcluster functions as a basic signaling unit during T cell activation[6,7]. Moreover, the dynamic interaction between TCR microclusters, actin cytoskeleton, and adhesion molecules leads to the formation of an immunological synapse between the T cell and APC to facilitate lysis of target cells, directed cytokine secretion, and other effector functions[3,8,9].

TCR microcluster formation is thought to involve noncovalent crosslinking between adapter and receptor proteins downstream of TCR ligation. One such mechanism involves cooperative interactions between LAT, SOS1, c-Cbl, and GRB2 molecules, in which multiple binding sites on LAT and SOS1 or c-Cbl for the SH2 and SH3 domains of GRB2 enable oligomerization of LAT-associated signaling molecules[10]. In similar fashion, oligomerization of the LAT signaling complex was shown to be induced by multivalent interactions between GADS, ADAP, SLP76, and LAT, suggesting that a combination of adapter interactions drives microcluster formation[11]. Expanding on the crosslinking model, an in vitro reconstitution study has proposed that microclusters form due to a phase transition mediated by crosslinked LAT, GRB2, and SOS1 molecules[12]. In addition, Lillemeier and colleagues have proposed a "protein island" mechanism, whereby TCR and LAT naturally exist as nanoscale clusters ('islands') and become linked to neighboring "islands" after TCR ligation to form microclusters[13].

Single molecule localization microscopy (SMLM) studies of activated T cells have produced different observations regarding the arrangement of TCR signaling molecules in microclusters. Studies proposing the "protein island" model have observed strong segregation between 50 and 200 nm-sized TCR and LAT "islands" while another study observed a relatively heterogenous mixture of nanoclusters and transient interactions of TCR and LAT[13,14]. These observed differences come as no surprise as SMLM images are prone to drift correction and registration errors[15,16], and can yield varying results depending on the reconstruction parameters[17,18]. In addition, SMLM studies utilize fixed cell samples and cannot observe the evolution of microcluster structures during T cell activation. To avoid these issues, we performed live, super-resolution, multicolor imaging of activated T cells using total internal reflection fluorescence structured illumination microscopy (TIRF-SIM).

Whereas the spatial localization of microcluster components has been studied in great detail, imaging recruitment kinetics has been challenging due to the rapid recruitment and internalization of signaling molecules at microclusters[19]. Moreover, kinetic studies of T cell activation have focused on the sequence of events downstream of microcluster formation such as vesicle recruitment, MTOC repositioning, and immune synapse formation[8,20–23]. The earliest study describing TCR-activated signaling microclusters reported that microcluster formation and calcium flux were detected immediately after contact between the T cell membrane and the anti-TCR-coated coverslip[6]. Extending these observations, Huse et al used photo-activatable pMHC to show that GRB2 clustering and calcium flux were observed at 4 s and 6–7 s after TCR stimulation[24].

Here we report that TCR microclusters contain two spatially distinct domains, and that these domains assemble in discrete, sequential kinetic steps. These kinetic steps are regulated by intracellular calcium flux dependent on T cell stimulation. The results suggest the presence of a previously unrecognized feedback mechanism that regulates T cell activation.

## Results

**Spatially segregated domains in TCR microclusters.** As some TCR microclusters are smaller than the diffraction limit (~250 nm) of conventional light microscopes, previous studies have utilized SMLM techniques to gain further insight into microcluster structure and composition. To date, SMLM studies have revealed patterns of molecular assembly within microclusters, the nanoscale clustering of molecular components, and the 3D distribution of concentrated receptors in resting T cells[13,14,25]. However, in addition to the limitations of SMLM mentioned above, we and others have reported artifactual clustering of photoactivatable fluorescent proteins used in SMLM[26,27] and poor resolution in antibody-based SMLM methods due to inadequate labeling of targeted structures[16,28,29]. Given these issues, we sought to use TIRF-SIM, which yields ~100 nm resolution, and is compatible with well characterized, monomeric fluorescent proteins to perform live imaging of microcluster formation in T cells[30].

Jurkat T cells expressing TCRζ-Halo conjugated to Janelia Fluor 646 ligand (JF646) and ZAP70-Emerald were activated on anti-CD3ε-coated glass and imaged using TIRF-SIM. TCRζ-Halo and ZAP70-Emerald signals co-localized at each microcluster (Fig. 1a) as expected from the known binding of ZAP70 to TCRζ ITAM residues. In contrast to TCRζ, LAT-Halo-JF646 and SLP76-Halo-JF646 were observed to be segregated from, but adjacent to ZAP70-Emerald in TIRF-SIM images of activated Jurkat cells (Fig. 1b, d). The same pattern was observed between endogenous TCRζ and LAT at individual microclusters (Fig. 1c). In activated Jurkat cells expressing GADS-Emerald, ZAP70-Apple, and GRB2-Halo-JF646, or LAT-Emerald, ZAP70-Apple, and SLP76-Halo-JF646, co-localization was observed between the adapter molecules (GADS and GRB2, LAT, and SLP76), but segregated from ZAP70 (Fig. 1e–f), showing that a sub-domain exists spatially distinct from TCR and ZAP70 within microclusters. Other LAT complex-bound signaling molecules ADAP, NCK, and VAV1 also localized to this segregated region adjacent

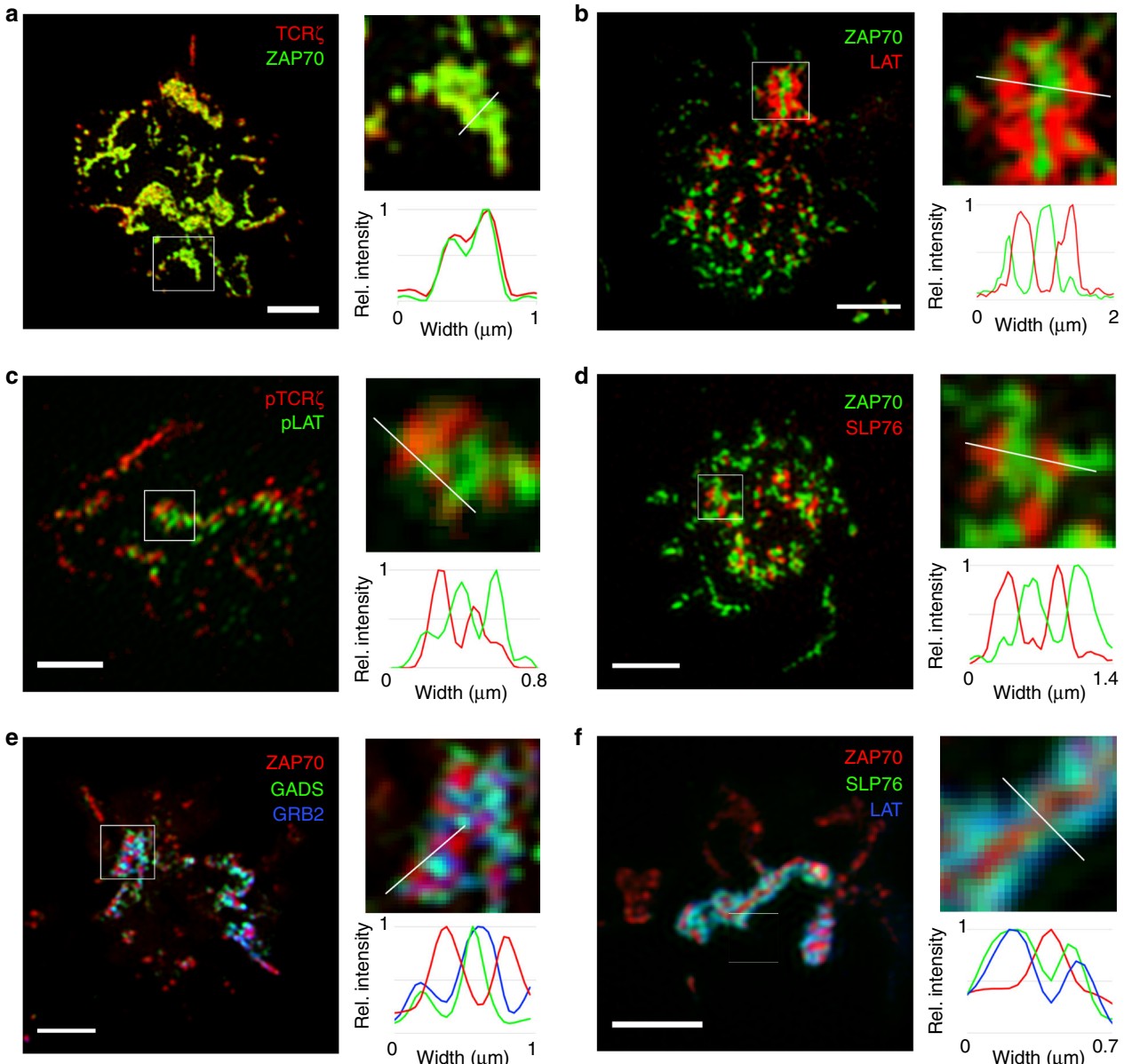

**Fig. 1** TIRF-SIM imaging of microclusters show distinct receptor and signaling domains. TIRF-SIM images of microclusters formed in Jurkat T cells activated on coverslip-bound anti-CD3 antibody were visualized using **a** TCRζ-Halo-JF646 (red) and ZAP70-Emerald (green), **b** ZAP70-Emerald (green) and LAT-Halo-JF646 (red), **c** anti-pTCRζ Y142; red) and anti-pLAT (Y226; green), **d** ZAP70-Emerald (green) and SLP76-Halo-JF646 (red), **e** GADS-Emerald (green), ZAP70-Apple (red), and GRB2-Halo-JF646 (blue), and **f** LAT-Emerald (blue), ZAP70-Apple (red), and SLP76-Halo-JF646 (green). **a–f** Upper right image was magnified from the region marked by a white box in the left image, and the bottom right graph shows relative (rel.) intensity measured across the width of the white line in the corresponding upper right image. **a–f** Scale bars in left images, 4 μm

to ZAP70 (Supplementary Fig. 1), indicating that the sub-domain represents the oligomerized LAT signaling complex. These observations show that two sub-diffraction-sized spatial domains exist within microclusters: one marked by TCR and ZAP70 and another by LAT-associated adapter and signaling molecules, which we will call the receptor and signaling domains.

**Stepwise recruitment of TCR microcluster components.** In addition to spatially distinct microcluster domains, the TIRF-SIM time-lapse images revealed a striking pattern in the dynamics of TCR microcluster formation. We consistently observed a delay in the appearance of ZAP70 at the newly formed, peripheral TCRζ clusters, and LAT or SLP76 at the peripheral ZAP70 clusters (Fig. 1a–d; Supplementary Movies 1–3). These observed non-

stochastic delays were surprising given the current "crosslinking" and "protein island" models which predict a stochastic process driven by interactions between membrane-bound molecular complexes. To better characterize the kinetics of molecular recruitment to TCR microclusters, we sought to slow microcluster formation by activating cells on anti-CD3ε-coated coverslips at room temp (21 ºC), and imaged cells using TIRF microscopy. As expected, activated Jurkat cells co-expressing CD3ε-YFP and TCRζ-Halo-JF646 showed simultaneous recruitment of both integral TCR components to microclusters (Fig. 2a, f [TCRζ-CD3ε]). In contrast, large delays were observed in TIRF time-lapse images between the recruitment of ZAP70-Apple to TCRζ-Emerald, Grb2-Emerald to ZAP70-Apple, and c-Cbl-YFP to ZAP70-Apple at individual microclusters (Fig. 2b–e). Quantification of the fluorescence intensities showed a kinetic lag of ~27 s

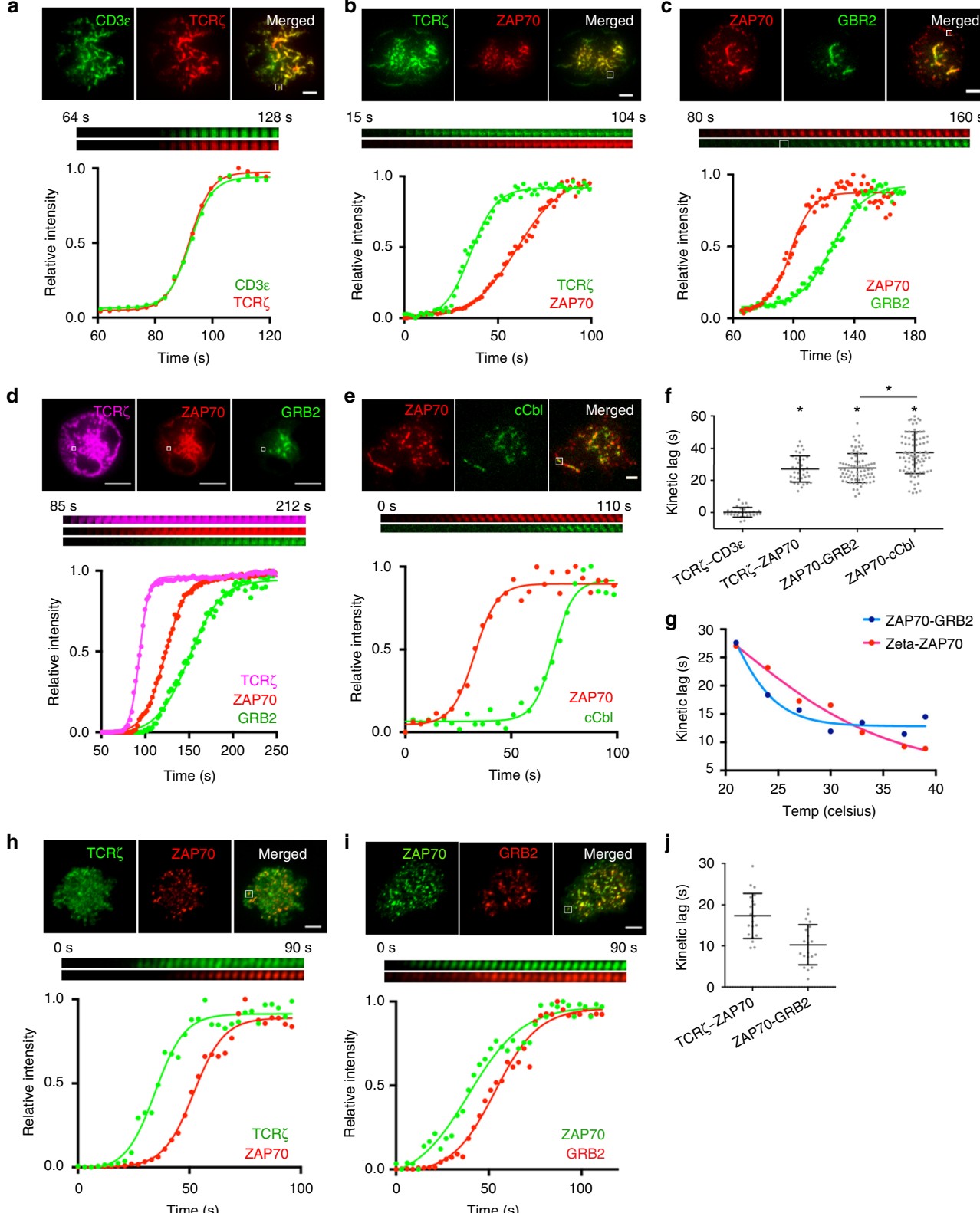

in both ZAP70 to TCRζ and GRB2 to ZAP70 recruitment and a larger kinetic lag of ~40 s in c-Cbl to ZAP70 recruitment (Fig. 2f).

Discrete, sequential kinetic lags were also observed in a Jurkat T cell triple-expressing TCRζ-Halo-JF646, ZAP70-Apple, and GRB2-Emerald (Fig. 2d). Switching the fluorescent proteins tags on ZAP70 and GRB2 did not affect their kinetic lag (Supplementary Fig. 2A, $p > 0.05$), and the kinetic lag between TCRζ-

GRB2 equaled the combined kinetic lags between TCRζ-ZAP70, and ZAP70-GRB2 (Supplementary Fig. 2B, $p > 0.05$) showing that the several lags in recruitment were not affected by the choice of tagged fluorescent protein or the combination of microcluster molecules used to measure it. The kinetic lags between TCRζ-ZAP70, and ZAP70-GRB2 were affected by temperature, though the ZAP70-GRB2 kinetic lag appeared to stabilize at ~15 s above

**Fig. 2** Microcluster components assemble in distinct sequential steps. TIRF images of microclusters formed in Jurkat T cells activated on coverslip-bound anti-CD3 antibody at 21 ºC were visualized using **a** CD3ε-YFP (green) and TCRζ-Halo-JF646 (red), **b** TCRζ-Emerald (green) and ZAP70-Apple (red), **c** ZAP70-Ruby (red) and GRB2-Emerald (green), **d** TCRζ-Halo-JF646 (magenta), ZAP70-Apple (red), and GRB2-Emerald (green), or **e** ZAP70-Apple (red) and c-Cbl-CFP (green). **a–e** Time-lapse montage at 3 s/frame (middle) and relative intensity plot (bottom) of the boxed region in the top image shows the kinetic lag between the recruitment of microcluster components. The bottom graph shows a plot of the relative intensity per acquired time frame as a colored circle and the best-fit sigmoidal curve as a colored line. Scales bars, 2 μm. **f** Average kinetic lag observed between the recruitment of respective microcluster components. Kinetic lag was measured by calculating the difference in half-max of best-fit sigmoidal curves. *$p < 0.0001$. **g** Kinetic lags measured between TCRζ-Emerald and ZAP70-Apple (red), or ZAP70-Apple and GRB2-Emerald (blue) in Jurkat cells activated on coverslip-bound antibody at indicated temperatures. **h–j** TIRF images of microclusters formed in primary human CD3[+] T cells activated on coverslip-bound anti-CD3 and anti-CD28 antibodies at 37 °C were visualized using **h** TCRζ-Emerald (green) and ZAP70-Scarlet (red), **i** ZAP70-Emerald (green) and GRB2-Scarlet (red). Time-lapse montage at 3 s/frame (middle) and relative intensity plot with best fit sigmoidal curve (bottom) of the boxed region in the top image. Scale bars, 2 μm. **j** Average kinetic lags (measured as above) observed between the recruitment of respective microcluster components

30 °C (Fig. 2g). The temp-dependence of TCRζ-ZAP70 and portions of ZAP70-GRB2 kinetic lags are consistent with the stochastic nature of protein-protein interactions. However, the temp-independence of the ZAP70-GRB2 kinetic lags at higher temp is surprising, and is possibly due to preferential clustering of LAT at specialized membrane domains,[31] or to the phase separation of crosslinked LAT oligomers[12], with the assumption that lipid domain segregation and phase separation occur selectively at higher temperatures. We note here the difference in the kinetic lags between TCRζ-ZAP70, and ZAP70-GRB2 at room temp (~27 s) and at 37 °C (~10 s), presumably the temperature at which most previous studies were done. Taken together, the above results provide evidence for a non-stochastic, stepwise recruitment of the kinase ZAP70 to the TCR, followed by recruitment of signaling proteins Grb2 and then c-Cbl to the ZAP70-bound TCR.

To verify that the stepwise recruitment of microcluster components were also observed in nontransformed cells, we adapted the assay to human primary T cells. CD3[+] cells were purified and expanded from human peripheral blood and transfected with either TCRζ-Emerald and ZAP70-Scarlet or ZAP70- Emerald and Grb2-Scarlet. Transfected cells were then dropped on stimulatory coverslips coated with anti-CD3 and anti-CD28 antibodies and imaged at 37 °C. As observed in Jurkat cells, primary human T cell blasts exhibited delays in the TIRF time-lapse images between both the recruitment of ZAP70-Scarlet to TCRζ-Emerald and Grb2-Scarlet to ZAP70-Emerald (Fig. 2h, i). Quantification of the fluorescence intensities showed a kinetic lag of ~17 s in ZAP70 recruitment to TCRζ and ~12 s in GRB2 to ZAP70 recruitment (Fig. 2j). Thus, although the kinetics are slightly different from what we observed in Jurkat cells, the stepwise recruitment of microcluster proteins was confirmed in primary human lymphoblasts at physiological temperature.

**Simultaneous recruitment of signaling domain molecules**. Upon TCR triggering, LAT phosphotyrosine motifs serve as binding sites for SH2 domain-containing proteins, including PLCγ1, Grb2, and Gads. Gads recruits the adapter SLP-76 to LAT and SLP-76 in turn recruits NCK, ADAP, and VAV1[1,2]. TIRF-SIM images showed that the signaling domain contains LAT and these LAT-associated adapters and enzymes. To test whether the signaling domain molecules follow the stepwise pattern observed between sequential recruitment of TCRζ, ZAP70, GRB2, and c-Cbl, we analyzed the kinetics of signaling domain molecule recruitment relative to ZAP70 or GRB2 at individual micro-clusters. In Jurkat cells co-expressing ZAP70-Apple and LAT-Emerald or SLP76-Emerald, the kinetic lags measured between the recruitment of LAT to ZAP70 and SLP76 to ZAP70 were of the same magnitude (~27) as between ZAP70-GRB2, suggesting that LAT, SLP76, and GRB2 are recruited simultaneously to ZAP70-marked microclusters (Fig. 3a–e; $p > 0.05$). Consistent

with the simultaneous recruitment of LAT, SLP76, and GRB2 adapter molecules, co-expression of PLCγ1-CFP, GADS-YFP, and GRB2-Apple or SLP76-CFP, NCK-YFP, and GRB2-Apple showed coincident appearance of PLCγ1, GADS, GRB2, SLP76, and NCK at individual microclusters (Fig. 3c, d). Moreover, in contrast to the large kinetic lags observed between ZAP70-GRB2 or GRB2-cCbl, little to no kinetic lags were observed between GRB2 and PLCγ1, GADS, NCK, SLP76, VAV1, or ADAP (Fig. 3e, $p < 0.05$ compared to ZAP70-GRB2 or GRB2-cCbl), indicating that LAT-associated adapters and enzymes become recruited to the signaling domain simultaneously. Overall, these results show that whereas signaling domain formation is delayed relative to the receptor domain, signaling domain components are recruited simultaneously during its formation, consistent with the highly cooperative nature of protein-protein interactions[10].

**Two different rates of signaling domain disassociation**. As T cell signals must be tightly regulated, we examined whether the signaling properties of microclusters were dynamically regulated by dissociation of signaling molecules. Previous studies have reported the rapid disappearance of signaling molecules at microclusters due to ubiquitinylation of LAT by the E3 ligase c-Cbl, which leads to internalization and degradation of the LAT signaling complex[19,32]. Consistent with these results, we observed a short kinetic lag between recruitment of Grb2 and c-Cbl that is visually apparent between the red (Grb2) and green (c-Cbl) curves during the recruitment phase (Fig. 4a left side of graph until ~100 s). Moreover, the peak in c-Cbl-YFP signal coincided with the peak of Grb2-Apple, following which a rapid decay in both signals were observed (Fig. 4a right side of graph after black arrowhead), suggesting that c-Cbl recruitment leads to signaling domain disassembly. Moreover, as the delayed recruitment of c-Cbl to GRB2-bound microclusters was shown here and in previous figures (Figs. 2e, f, 3e), the GRB2-c-Cbl kinetic lag likely represents the time window between active signaling and signal attenuation at individual microclusters. It is worth noting here that the kinetic lag between ZAP70-c-Cbl (Fig. 2e, f) is equivalent to the combined lags between ZAP70-GRB2 and GRB2-c-Cbl (Fig. 3e).

Whereas the signaling domain molecules appeared simultaneously at individual microclusters, we detected differences in their rates of disassociation. Differences in the rates of dissociation were observed between the three major signaling molecules bound to LAT phosphotyrosines: Gads, Grb2 and PLCγ1. While GADS-YFP signal decreased rapidly, PLCγ1-CFP and Grb2-Apple signals showed slower dissociation kinetics (Fig. 4b). Similar to Gads, faster decay in signals of the Gads-bound protein SLP-76, and SLP-76-bound NCK and VAV1 was observed indicating that signaling domain molecules undergo fast and slow rates of disassociation from microclusters (Fig. 4c, d). The difference in dissociation rates could result from the

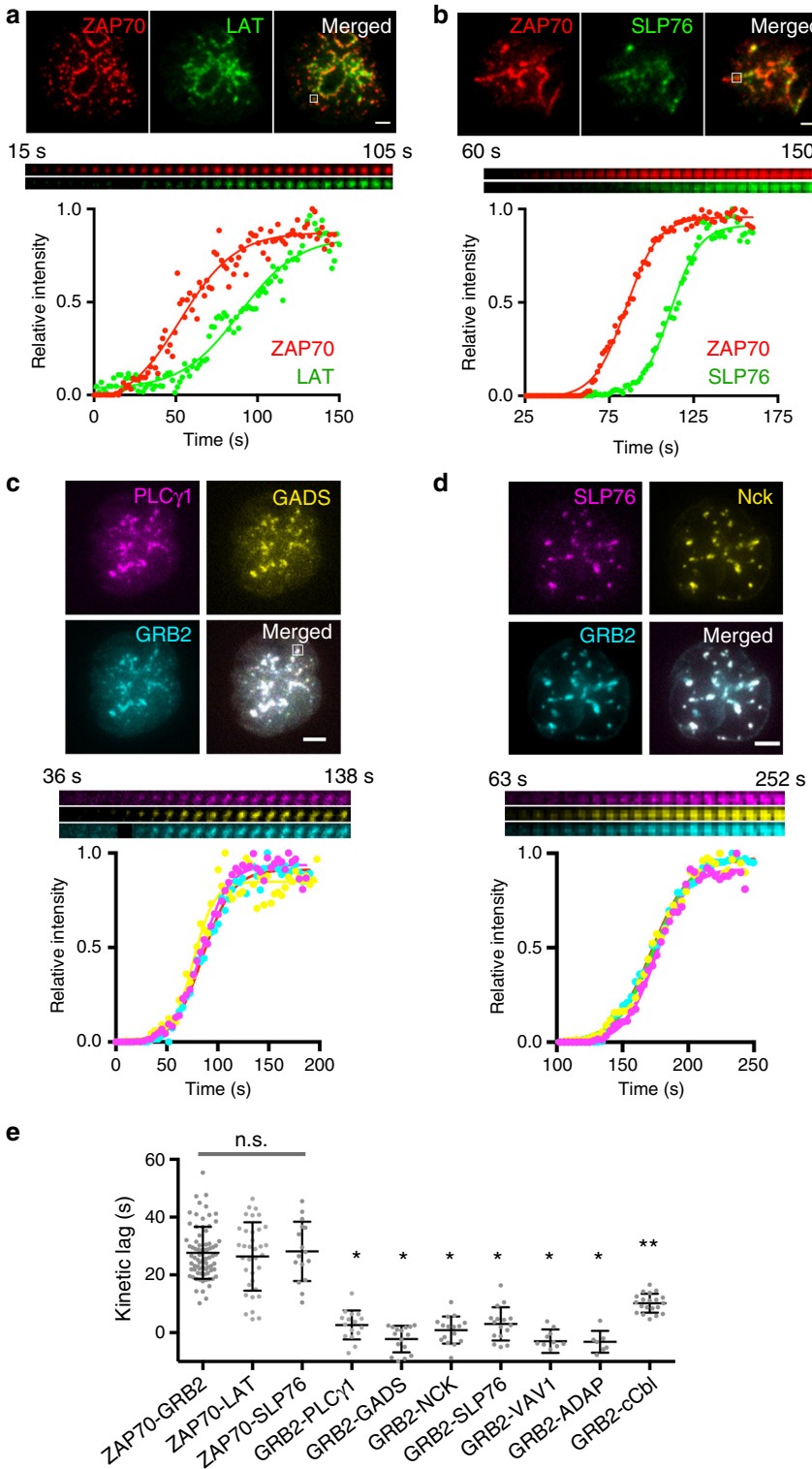

**Fig. 3** Signaling cluster components are recruited simultaneously to microclusters. TIRF images of microclusters formed in Jurkat T cells activated on coverslip-bound anti-CD3 antibody at 21°C were visualized using **a** LAT-Emerald (green) and ZAP70-Apple (red), **b** SLP76-Emerald (green) and ZAP70-Apple (red), **c** PLCγ1-CFP (magenta), GADS-YFP (yellow), and GRB2-Apple (cyan), or **d** SLP76-CFP (magenta), NCK-YFP (yellow), and GRB2-Apple (cyan). **a–d** Time-lapse montage (middle) and relative intensity plot (bottom) of the boxed region in the top image shows simultaneous recruitment of signaling cluster components. The bottom graph shows a plot of the relative intensity per acquired time frame as a colored circle and the best-fit sigmoidal curve as a colored line. Scales bars, 2 μm. **e** Average kinetic lag observed between the recruitment of indicated pair of microcluster components at 21 °C. n.s., not significant. *$p < 0.0001$ compared to ZAP70-GRB2. **$p < 0.0001$ compared to kinetic lags from GRB2-PLCγ1 to GRB2-ADAP

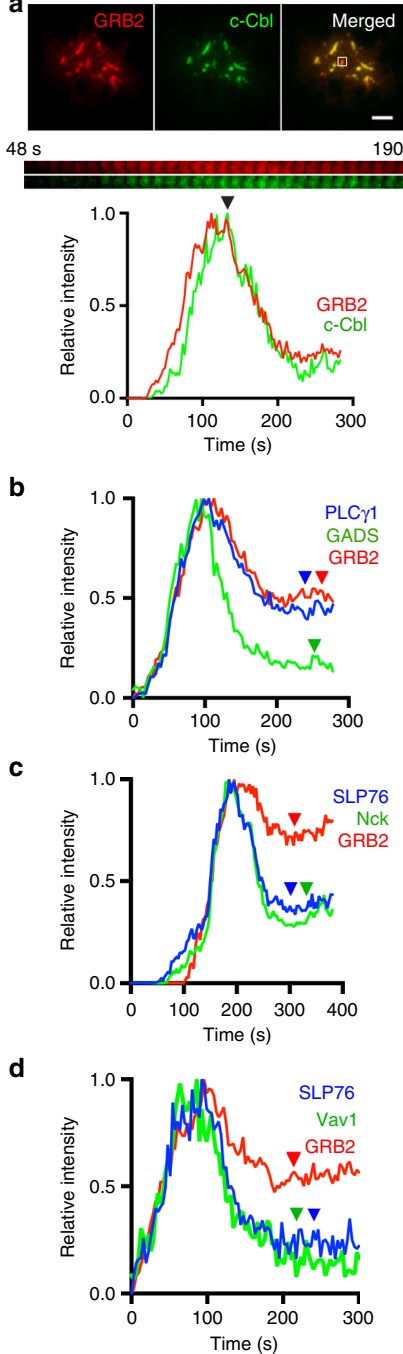

**Fig. 4** Signaling domain disassociation coincides with c-Cbl recruitment and occurs at two distinct rates. **a** TIRF image of microclusters formed in Jurkat T cells activated on coverslip-bound anti-CD3 antibody at 21 °C were visualized using c-Cbl-YFP and GRB2-Apple. Time-lapse montage (middle) and relative intensity plot (bottom) of the boxed region in the top image shows decrease in GRB2 and c-Cbl signal coincident with peak c-Cbl recruitment (black arrowhead). Scale bar, 2 μm. **b–d** Representative relative intensity plots of **b** PLCγ1-CFP (blue), GADS-YFP (green), and GRB2-Apple (red), or **c** SLP76-CFP (blue), NCK-YFP (green), and GRB2-Apple (red), or **d** SLP76-CFP (blue), Vav1-YFP (green), and GRB2-Apple (red) at microclusters. The colored arrowheads indicate the final intensity levels after disassociation of each corresponding signaling cluster component

accelerated removal of the SLP76 complex from microclusters due to pulling forces from the attached actin network[33,34], differential rates of ubiquitinylation or varying rates of tyrosine dephosphorylation by different sets of phosphatases.

**Signaling domain recruitment leads to effector function**. The LAT signaling complex is required for amplification of TCR signaling and initiation of downstream T cell activation pathways including intracellular calcium flux and actin polymerization-driven cell spreading[1]. To observe the sequence of downstream TCR signaling steps relative to microcluster formation, we measured total signal intensity at the synapse in a Jurkat T cell loaded with Fluo-4-AM, and expressing ZAP70-Apple and, as a component of the signaling domain, GRB2-Halo. We observed stepwise recruitment of ZAP70-Apple and GRB2-Halo-JF646 to microclusters, followed by initiation of calcium flux (Fig. 5a, Supplementary Movie 4). We note here the sharp vertical spike in intracellular calcium flux amid the gradual increase in GRB2 intensity, which likely reflects crossing of a critical threshold of signaling complex recruitment required to initiate the calcium flux event. Likewise, initiation of rapid cell spreading was observed after the recruitment of GRB2 to microclusters, shown by the kymograph of an activated cell expressing Zeta-Halo-JF646 and GRB2-Emerald (Fig. 5b, Supplementary Movie 5). Previous studies examining LAT signaling complex function have utilized the LAT-deficient J.CaM2.5 cell line, which could harbor additional genetic changes from chemical mutagenesis, and has been reported to contain residual expression of LAT[19,35]. Therefore, we sought to re-examine calcium flux and cell spreading, and to analyze the kinetics of microcluster formation in a CRISPR-generated LAT knockout Jurkat cell line, J.LAT KO[36]. The absence of LAT in J.LAT KO cells was confirmed using western blot and immunofluorescence (Supplementary Fig. 3A,C). Importantly, calcium flux was not detected in activated J.LAT KO cells, phenocopying J.CaM2.5 cells (Supplementary Fig. 3B). However, whereas phospho-LAT (pLAT) signal was not observed in stimulated J.LAT KO cells, low amounts of pLAT signal persisted in J.CaM2.5 cells (Supplementary Fig. 3C), validating J.LAT KO cells for functional studies of LAT.

Though TCRζ, ADAP, SLP76, GADS, VAV1, PLCγ1, NCK, GRB2, and c-Cbl localized to microclusters in E6.1 Jurkat cells, only TCRζ, GRB2, NCK, and c-Cbl signals were apparent at ZAP70-marked microclusters in J.LAT KO cells, with a partial localization for NCK (Fig. 5c). The residual localization of GRB2, c-Cbl, and NCK at microclusters in the absence of LAT is perhaps due to their known interaction with TCRζ, ZAP70 and CD3ε[37–39]. Despite the lack of a functional signaling domain, as defined above, TCRζ-ZAP70, ZAP70-GRB2, and GRB2-cCbl kinetic lags were unaffected in J.LAT KO cells (Fig. 5d–g, $p < 0.05$ compared to E6.1). Given that multivalent crosslinking of LAT is thought to drive overall microcluster formation[10,12], these results are somewhat surprising, and indicates that receptor domain formation can be independent of LAT crosslinking. In addition, molecules that can be directly recruited to proteins in the receptor domain (GRB2, c-Cbl, and NCK) can bypass the LAT cross-linking mechanism.

The WASP family of proteins play an important role in bridging TCR signaling to F-actin polymerization[33]. In particular, the WASP family member, WAVE/Scar is critical for ARP2/3-driven F-actin polymerization downstream of TCR activation, leading to rapid, radial lamellipodial growth in T cells[33,40,41]. Of the three isoforms, we chose WAVE1 for its selective localization to the leading edge of lamellipodia[42] to report ARP2/3-driven cell

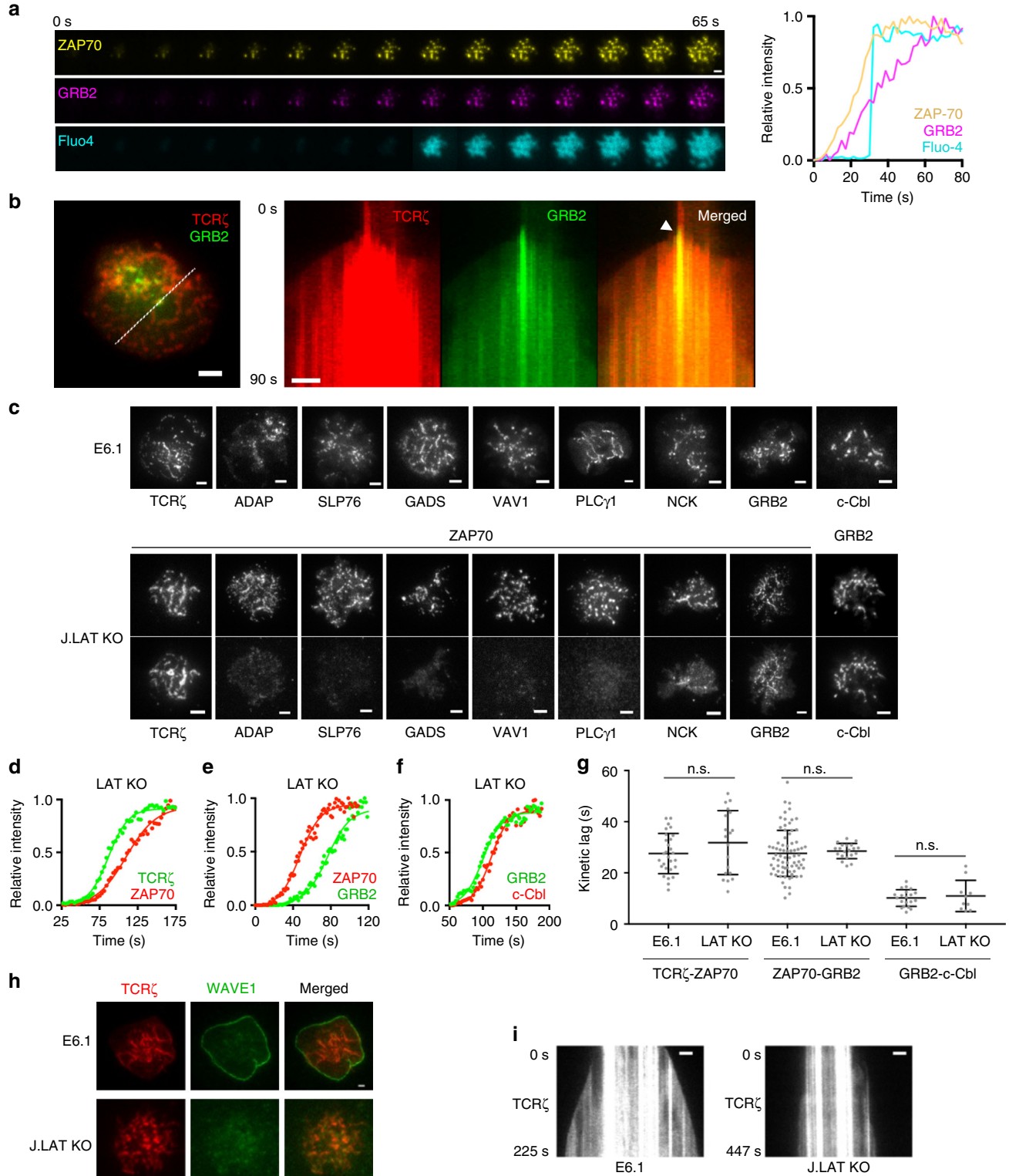

expansion in activated T cells. Whereas mEmerald-WAVE1 localized strongly to the leading edge of stimulated E6.1 cells, WAVE1 was not detected at the cell periphery in J.LAT KO cells (Fig. 5h), consistent with the key requirement of the LAT complex in F-actin polymerization-driven cell spreading. Additionally, rapid cell spreading was not observed in stimulated J. LAT KO cells (Fig. 5i), despite the normal recruitment kinetics of TCRζ, ZAP70, GRB2, and c-Cbl to microclusters (Fig. 5g). Together these results show that LAT is required for the

formation of a functional signaling domain, and for progression into downstream T cell activation pathways.

**Kinetic lag in microcluster recruitment is calcium-dependent.**
T cell activation begins with the formation of microclusters at focal contacts between T cell membrane protrusions and the activating surface, and is sustained by continued formation of microclusters at the periphery of the spreading cell[6,7]. As the early and late-formed microclusters differ both in geography and

**Fig. 5** Signaling cluster recruitment is required for progression to downstream T cell activation events. **a** Time-lapse montage (left) of a Jurkat T cell becoming activated on coverslip-bound anti-CD3 antibody at 21 °C expressing ZAP70-Apple (yellow), GRB2-Halo-JF646 (magenta), and Fluo-4 (cyan), and the corresponding intensity plot over time (right) shows that downstream calcium flux follows recruitment of signaling cluster as reported by GRB2-Halo-JF646. **b** Right, kymograph of TCRζ-Halo-JF646 and GRB2-Emerald in an activating Jurkat T cell corresponding to the white line in the left TIRF image shows that cell spreading initiates after recruitment of signaling cluster (white arrowhead in merged image). **c** Representative TIRF images of TCRζ-Emerald, ADAP-GFP, SLP76-CFP, GADS-YFP, VAV1-YFP, PLCγ1-CFP, NCK-YFP, GRB2-Emerald, c-Cbl-CFP localized in activated E6.1 Jurkat cells (top) or with either ZAP70-Apple or GRB2-Apple in activated J.LAT KO (LAT$^{-/-}$) Jurkat cells. Representative relative intensity plots of **d** TCRζ-Emerald (green) and ZAP70-Apple (red), **e** ZAP70-Apple (red) and GRB2-Emerald (green), or **f** c-Cbl-YFP and GRB2-Apple at microclusters and **g** average kinetic lag measured between indicated pair of microcluster components in J.LAT KO cells. **h** Activated E6.1 and J.LAT KO Jurkat cells co-expressing TCRζ-Halo-JF646 and mEmerald-WAVE1. **i** Representative kymographs of TCRζ-Emerald in an activated E6.1 and J.LAT KO Jurkat cell. Scale bars, 2 μm

function, we sought to analyze the kinetic lags between molecular recruitment in both early and late-formed microclusters. TIRF imaging of a Jurkat T cell expressing TCRζ-Emerald and ZAP70-Apple was initiated before the cell made contact with the anti-CD3ε-coated coverslip and imaged for 4 min to capture the full temporal spectrum of microcluster formation. Strikingly, we observed little to no kinetic lag between TCRζ-ZAP70 in the earliest formed microclusters (Fig. 6a, left, 50–100 s), with a transition to longer kinetic lags as T cell activation progressed (Fig. 6a, right, compare 75 s [early] to 150 s [late]). We note that the kinetic lags in late-formed microclusters settled in a range between 20 and 40 s with an average of ~27 s, matching the TCRζ-ZAP70 kinetic lags measured previously (compare 100 and 200 s in Figs. 2f to 6a) and indicating that kinetic lag measurements in previous figures reflect late-formed microclusters. The same pattern was observed for ZAP70-GRB2 kinetic lags, with little to no kinetic lag observed between recruitment of GRB2 to ZAP70 in early formed microclusters (Fig. 6b, left, 0–50 s), increasing kinetic lag over time (Fig. 6b, right, compare 34 s [early] to 110 s [late]), and kinetic lags in late-formed microclusters matching ZAP70-GRB2 kinetic lags measured in previous figures (compare 50–150 s in Figs. 2f to 6b).

Curiously, we noticed that the transition to longer kinetic lags coincided in time with the calcium flux event observed in Fig. 5a. As calcium plays an important and diverse role as a second messenger system in cells, and as intracellular calcium levels shift drastically due to TCR stimulation, we sought to test whether intracellular calcium affects microcluster recruitment kinetics in activated T cells. Indeed, calcium chelation using EGTA and BAPTA significantly decreased the kinetic lags between TCRζ-ZAP70 and ZAP70-GRB2 in late-formed microclusters (Fig. 6c, d at 21 °C and Supplementary Fig. 4A at 37 °C), indicating that microcluster recruitment kinetics are calcium-dependent. Over-expression of the genetically encoded calcium reporter, GCaMP6 phenocopied the effect of EGTA/BAPTA on kinetic lags (Fig. 6c, d), likely due to the chelating effect of its calcium-binding calmodulin domain[43]. Importantly, increasing the extracellular concentration of calcium rescued the phenotype. When GCaMP6 over-expressing cells were activated in a 5 mM CaCl$_2$ solution, instead of 2 mM CaCl$_2$, the TCRζ-GRB2 and ZAP70-GRB2 kinetic lags in late-formed microclusters were unchanged from DMSO-treated cells (Fig. 6c, d). To evaluate the effects of calcium increases in microcluster kinetics, cells were treated with the calcium ionophore ionomycin immediately prior to being dropped on coverslips. Converse to the effects of calcium chelation, calcium elevation led to a significant increase in kinetic lags (Fig. 6d). Lastly, to confirm the correlation between microcluster recruitment kinetics and calcium concentration, we titrated the concentration of extracellular calcium in the imaging buffer and measured the kinetics lags at each concentration. Indeed, the kinetic lags in late-formed microclusters correlated with extracellular calcium concentration, as TCRζ-GRB2 and ZAP70-GRB2 kinetic lags were short at low

concentrations (0–0.5 mM CaCl$_2$) and longer at physiological concentrations (1.0–2.0 mM CaCl$_2$), with the transition to longer kinetic lags taking place between 0.5 and 1.0 mM CaCl$_2$ (Fig. 6e, f). Taken together, the results in Fig. 6 show that the kinetic delays between the stepwise recruitment of microcluster components is dependent on calcium and is regulated by intracellular calcium flux downstream of TCR activation.

## Discussion

Super-resolution imaging of activated Jurkat T cells revealed that microclusters contain spatially distinct receptor and signaling domains at the sub-diffraction scale. This spatial separation of microcluster compartments is supported by previous biochemical studies showing that LAT molecules are corralled within distinct membrane domains[31,44], and is visually consistent with the segregated TCR and LAT islands observed in SMLM studies[13]. However, the stepwise, sequential kinetics of receptor domain clustering, ZAP70 kinase recruitment, and signaling domain formation, which we detect, is contrary to the stochastic linking of TCR and LAT islands predicted by the "protein island" model. Furthermore, whereas the proposed sizes of islands (50–200 nm) or preformed clusters[45] are within the resolution limit of TIRM-SIM (80–120 nm), preformed TCR and LAT clusters were not observed in resting cells or during the early stages of T cell activation. In contrast, other super-resolution studies have reported significantly smaller sizes for TCR and LAT complexes, as oligomers of 2–5 TCR and LAT molecules[14], or strictly as monomeric TCRs[46]. As such nano-scale sizes are beyond the resolution limit of TIRF-SIM, imaging techniques with higher resolution will need to be utilized to further investigate the nano-scale arrangement of molecular components within the receptor and signaling domains, such as correlative super-resolution imaging using both TIRF-SIM and SMLM.

Spatial segregation between receptor and signaling domains is reminiscent of secretion and signaling domains reported in a CTL synapse[22]. However, the domains described here are at smaller length scales and occur in seconds rather than minutes after activation. The spatial separation and the kinetics of molecular recruitment of the receptor and signaling domains we observed suggest distinct molecular mechanisms driving the oligomerization of TCR and LAT complexes. Our results showed that TCR clustering and ZAP70 recruitment to TCR were independent of LAT. This is possibly due to crosslinking from the coverslip-bound anti-TCR antibody, or to self-oligomerization of ligand-bound TCRs[47]. In addition, as cooperativity of defined protein–protein interactions drives the oligomerization of LAT complexes, the simultaneous recruitment of signaling domain molecules is consistent with stochastic crosslinking of multi-protein complexes[10,11]. However, two key questions are raised regarding the segregated but neighboring microcluster domains. First, if the signaling domain is composed of LAT-associated complexes and is formed by their mutual crosslinking, why does

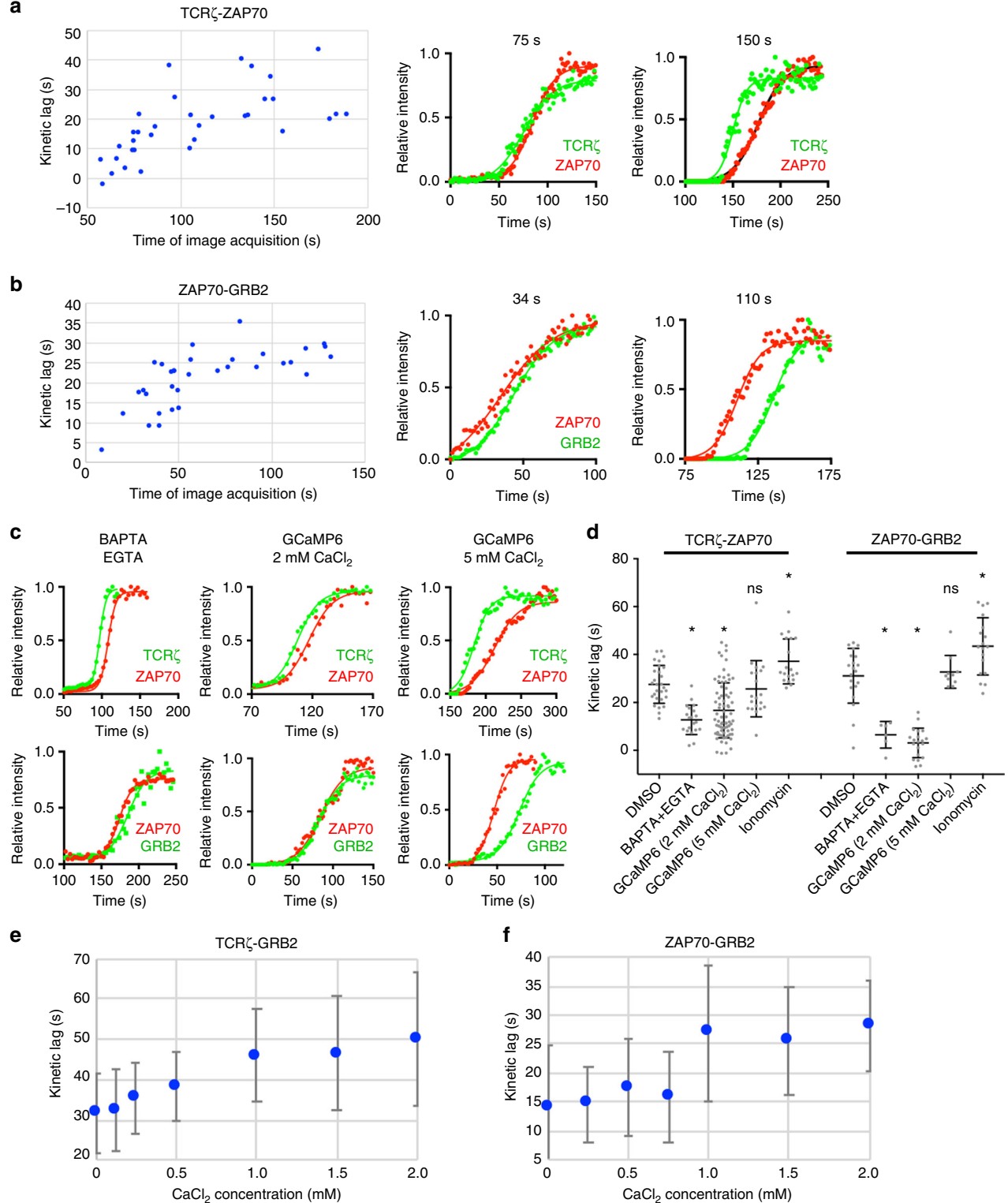

signaling domain formation take place directly adjacent to the receptor domain? One possibility is that the ZAP70 kinase is activated while tethered to the TCR within the receptor domain, becomes detached and phosphorylates substrates in nearby domains as described previously[48]. However, the released pool of active ZAP70 will soon encounter phosphatases surrounding the microcluster and become inactivated. Thus, substrates such as LAT and SLP76 are required to be in proximity of the receptor domain to be robustly phosphorylated by activated ZAP70. In

support of this idea, we note that the previously reported length scales of ZAP70 movement at the membrane (~100 nm) correlates to distances between the receptor and signaling domains[48].

Second, what is the physical connection between the two domains? As the Nck-bound LAT complex is known to interact with the highly dynamic actin cytoskeleton[33], molecular binding between the receptor and signaling domain components would need to counteract the pulling forces from the inward actin flow. Indeed, over time a portion of peripheral signaling domain

**Fig. 6** Kinetic lags between recruitment of microcluster components are dependent on intracellular calcium flux. **a** Plot of kinetic lags measured between TCRζ-Emerald and ZAP70-Apple in microclusters formed at indicated image acquisition time in the same Jurkat T cell activated at 21 °C (left). Representative intensity plots of TCRζ-Halo-JF646 (green) and ZAP70-Emerald (red) measured in a microcluster at 75 s and 150 s after T cell activation (right). **b** Plot of kinetic lags measured between ZAP70-Emerald and GRB2-Halo-JF646 in microclusters formed at indicated image acquisition time in the same Jurkat T cell activated at 21 °C (left). Representative intensity plots of ZAP70-Emerald (red) and GRB2-Halo-JF646 (green) during early (34 s, middle) and late (110 s, right) temporal phases of T cell activation (right). **c** Representative intensity plots of TCRζ-Emerald (green) and ZAP70-Scarlet (red) (top row), or ZAP70-Scarlet (red) and GRB2-Emerald (green) (bottom row) at a microcluster in a Jurkat cell treated with BAPTA and EGTA (left column), or between TCRζ-Halo-JF646 (green) and ZAP-Scarlet (red) (top row), or ZAP70-Scarlet (red) and GRB2-Halo-JF646 (green) (bottom row) in GCaMP6-expressing cells in an imaging buffer with 2 mM $CaCl_2$ (middle column), or 5 mM $CaCl_2$ (right column). **d** Average kinetic lags measured between TCRζ and ZAP70, or ZAP70 and GRB2 at microclusters under the indicated calcium chelation or elevation conditions. *$p < 0.0001$ compared to DMSO. n.s., not significant compared to DMSO. Average kinetic lags measured between **e** TCRζ-Emerald and GRB2-Scarlet, or **f** ZAP70-Emerald and GRB2-Scarlet at microclusters at indicated extracellular $CaCl_2$ concentrations

clusters succumbed to the pulling force, and became detached from microclusters as evident from time-lapse TIRF-SIM images (Supplementary Fig. 5, Supplementary Movie 6). However, most signaling domain clusters remained adjacent to ZAP70-marked receptor domains and remained immobile as shown in Fig. 1, implying a connection between the two spatially distinct domains. One possible connection is an adapter molecule, an example of which is Shb, which has been shown to bind TCRζ and LAT, GRB2, and other signaling molecules[49–51]. Another intriguing possibility is the Src-family kinase Lck, shown recently to play a scaffolding role by binding both ZAP70 and LAT[36]. Such a role for either Shb or Lck could be confirmed by super-resolution imaging, with their localization expected between the receptor and signaling domains.

Contrary to the stochastic enlargement of microclusters predicted by the "crosslinking" and "protein island" models, we observed a non-stochastic process of microcluster assembly through stepwise recruitment of TCR, ZAP70, LAT-associated signaling complex, and c-Cbl. As the stepwise recruitment pattern follows the known biochemical sequence of TCR signaling, and as disruption of signaling domain recruitment impeded the progress of downstream pathways in J.LAT KO cells, our kinetic results have multiple implications for models of TCR signal transduction such as the kinetic proofreading model[52,53]. First, whereas the kinetic proofreading model predicts the discrimination of pMHCs based on the probability of a sequence of contingent TCR modifications occurring (i.e., ZAP70 binding to the phosphorylated TCRζ) before the ligand disassociates from TCR, the large kinetic delays between the ensemble recruitment of ZAP70 to TCRζ and the LAT complex to ZAP70 may need to be considered when calculating the kinetic parameters. In particular, the long, calcium-dependent delay in the recruitment of ZAP70 to TCR clusters (~27 s at 21 °C, ~10 s at 37 °C) is surprising given the high affinity of ZAP70 for phosphorylated ITAMs[54]. In addition to the stochastic inhibition from phosphatase activity, and molecular competition for ITAM binding, this non-stochastic, calcium flux-induced kinetic delay may indicate a novel negative feedback mechanism to prevent ZAP70 recruitment to activated TCR.

Second, the kinetic delays between recruitment of microcluster components has implications beyond the scope of the kinetic proofreading model. For instance, the delayed recruitment of c-Cbl relative to the signaling domain, which represents a time window of active signaling before signaling domain disassociation, is not included in such T cell activation models[52]. As the functional signaling domain is required for downstream signaling events such as calcium flux and actin polymerization-driven cell spreading, and as the kinetic lags are modulated by calcium-dependent mechanisms, this active signaling window is a potential target for downstream regulation of TCR signaling.

Third, our results show that calcium signaling has an inhibitory effect on TCR signaling. As the induction of calcium flux was shown to significantly increase kinetic lags, and as the kinetic proofreading model predicts a correlation between activation threshold for TCR ligand interactions and kinetic delays between the signaling steps[53], TCR signaling would be dampened by intracellular calcium flux. In support of this, a previous study showed increased phosphorylation of TCR signaling components and enlargement of TCR microclusters after calcium chelation[20]. On the other hand, artificially increasing calcium in Jurkat cells promoted actin polymerization and reduced TCR mobility on the T cell surface[55], suggesting that calcium flux may alter TCR microcluster kinetics through manifold pathways. Overall, our results point to an important distinction between TCR activation thresholds before and after calcium flux, and to a new role for calcium in downregulating TCR signaling. Additionally, whereas previously reported negative feedback mechanisms have focused on the role of phosphatases such as SHP-1 in reversing key phosphorylation events during TCR activation[56,57], the calcium-dependent kinetic delays in microcluster assembly represents a potentially new mode of TCR regulation.

We note here that the spatial and kinetic results were obtained using a specific in vitro system at room temp to study T cell activation. To further substantiate these findings, microcluster kinetics should be probed under other activation settings. Furthermore, future experiments should test the effect of agonist strength on kinetic recruitment parameters. Having stated this, we reiterate the strength of our in vitro system, which allowed consistent imaging of TCR activation with high spatial and temporal resolution. Whereas activating T cells at room temp revealed the large kinetic lags in microcluster recruitment, significant kinetic lags were observed at physiological temp as shown in Fig. 2g. Additional experiments will be needed in the future to extend our results and to elucidate the molecular mechanism of calcium-dependent kinetic delays during the stepwise assembly of TCR microclusters.

## Methods

**DNA constructs.** Constructs used in this paper were generated as follows: LAT-Emerald, GRB2-Emerald, ZAP70-Emerald, SLP76-Emerald, GADS-Emerald, and TCRζ-Emerald were generated by cloning AgeI-NotI (LAT-Emerald, ZAP70-Emerald, TCRζ-Emerald) or AgeI-BsrGI (Grb2-Emerald) or AgeI-NheI (SLP76-Emerald) digested Emerald sequence from Emerald-zyxin6 (Addgene, plasmid no. 54319) into plasmids containing the respective cDNA sequences. Generation of ZAP-Apple and Grb2-Apple were described previously[20]. ZAP70-Scarlet and GRB2-Scarlet were generated by cloning ZAP-70 or Grb2 cDNAs into HindIII-BamHI or NheI-AgeI digested pLifeAct_mScarlet_N1 plasmid respectively (Addgene, plasmid# 85054). TCRζ-Halo, LAT-Halo, SLP76-Halo, GRB2-Halo, and ZAP70-Halo were generated by cloning PvuI-SbfI (LAT-Halo, ZAP70-Halo, Grb2-Halo) or NheI-ApaI (Grb2-Halo, SLP76-Halo) digested Halo sequence from pHTC-Halotag CMV-Neo vector (Promega) into plasmids containing the respective cDNA sequences. mEmerald-WASP1-N-14 was a gift from Michael Davidson (Addgene plasmid # 54315). pGP-CMV-GCaMP6m (GCaMP6) was a gift from Douglas Kim (Addgene plasmid # 40754). Generation

of Nck-GFP, Nck-YFP, ADAP-GFP, Vav1-GFP, Vav1-YFP, c-Cbl-YFP, c-Cbl-CFP, PLCγ1-CFP, CD3ε-YFP, GADS-YFP, and SLP76-CFP were described previously[6,33].

**Reagents**. Human anti-CD3 (HIT3a clone) monoclonal antibodies were purchased from Pharmingen and were used to coat coverslips for imaging assays and OKT3 (produced in the laboratory) was used to trigger T cell activation in calcium flux assayed by flow. The following antibodies were used for immunostaining: mouse anti-pLAT[226] (BD Biosciences, 558363) and mouse monoclonal anti-TCRζ (pY142) antibody (BD Biosciences, 558402). BAPTA-AM, Indo-1-AM and Fluo-4-AM were from Invitrogen. EGTA and ionomycin were purchased form Sigma. Halo-tag ligand conjugated to Janelia Fluor (JF)-646 and -549 were gifts of Luke Lavis at Janelia Research Institute.

**Cell culture and transfection of Jurkat cells**. Jurkat E6.1 cells have been described previously[32]. J.LAT KO and J.CaM2.5 cells were gifted from Art Weiss and have been described previously[36]. E6.1 Jurkat cells were cultured in RPMI (11875–093; Life Technologies), 10% fetal bovine serum (26140–079; Life Technologies), and 1% penicillin–streptomycin (Life Technologies). For transient transfections, 1e6 Jurkat cells were transfected with 2 μg DNA using the Nucleofector Kit V (Lonza, catalog no. VCA-1003), Program X-001 24 h prior to imaging. Prior to imaging Jurkat cells were spun down and resuspended in imaging buffer (20 mM Hepes pH 7.2, 137 mM NaCl, 5 mM KCl, 0.7 mM Na$_2$HPO$_4$, 6 mM D-glucose, 2 mM MgCl$_2$, 2 mM CaCl$_2$, 1% BSA). For calcium chelation and calcium titration experiments imaging buffer without CaCl$_2$ was used.

**Cell culture and transfection of primary human T cells**. Primary human peripheral blood T cells were obtained without donor identifiers from the National Institutes of Health Blood Bank. Mononuclear cells were isolated by Ficoll density gradient centrifugation. Untouched CD3$^+$ cells were isolated using the MACS Pan T Cell Isolation Kit (130–096–535; Miltenyi Biosciences). Lymphoblasts were generated by activation with human T cell TransAct (130–111–160; Miltenyi Biosciences) in RPMI supplemented with 10% FBS, 1% penicillin–streptomycin and 50 U/ml of human IL-2. T lymphoblasts were cultured at 37 °C in 5% CO$_2$ for 6 days in exponential growth phase. Cells were then cultured for an additional day in the presence of 10 U/ml of IL-2. Cells were transfected with a LONZA electroporator using LONZA solution for primary human T cells (VPA-1002; LONZA) and protocol T-23. Primary human T cells were imaged 5 h after transfection.

**Live cell imaging**. Preparation of antibody-coated coverslip has been described previously[58]. Jurkat cells were added to HIT3a antibody-coated 8-well coverslip chambers (Lab-Tek, Thermo Fisher) and imaged at RT (21 °C) or other specified temperature in imaging buffer. Jurkat cells expressing Halo-tagged constructs were labeled with 100 nM Janelia Fluor Halo-647 ligand for 30 min at 37 °C. Cells were imaged after three washes in complete medium and then resuspended in imaging buffer. All chambers used for live cell imaging contained imaging buffer. TIRF images from live cells were collected with a Nikon Ti-E inverted microscope, using a 100× SR Apochromat TIRF objective lens (1.49 numerical aperture), and an Andor iXon Ultra 897 EM charge-coupled device camera (512 × 512 pixels, 16 μm pixel). Time-lapse images were collected at 3 s/frame except for experiments with short kinetic lags such as Figs. 3e and 4, which were collected at 1.5 s/frame. The TIRF-SIM microscope used in these experiments is housed in the Advanced Imaged Center (AIC) at the Howard Hughes Medical Institute Janelia research campus. The system is configured and operated as previously described[30]. In brief, images were acquired using an inverted microscope (Axio Observer; ZEISS) fitted with a 100 × 1.49 NA objective (Olympus) and a spatial light modulator to provide structured illumination. TIRF-SIM image in Fig. 1c was collected using the Delta Vision OMX SR microscope using the 2D-SIM TIRF mode (GE Healthcare). TIRF-SIM time-lapse images were collected at 1 s/frame and at 37 °C for Fig. 1a–c and at 21 °C for Fig. 1d–f and Supplementary Fig. 1. For calcium chelation experiments, Jurkat cells were preloaded with BAPTA-AM (20 μM) for 20 min at 37 °C in complete media buffered with 1 mM EGTA. BAPTA-loaded cells were rinsed and pipetted into coverslip chambers containing imaging buffer supplemented with 1 mM EGTA. To increase intracellular calcium levels preceding TCR stimulation, Jurkat cells were pretreated with ionomycin (1 μM) immediately prior to being pipetted into coverslip chambers. Calcium flux was imaged using Jurkat cells loaded with Indo-1-AM for flow cytometry and Fluo-4-AM for TIRF microscopy. For TIRF imaging of early formed microclusters in Fig. 6, expression of fluorescently tagged protein in cells was gauged by epifluorescence before the cells made contact with the anti-TCR-coated coverslip surface. For cells with the appropriate level of fluorescent signal TIRF imaging was initiated before the cells made with the coverslip in anticipation of early microcluster formation.

**Immunofluorescence imaging**. Jurkat T cells were allowed to adhere to the HIT3a antibody-coated coverslip for 2 min at 37 °C in imaging buffer and then fixed for 30 min in 4% (wt/vol) PFA solution (Electron Microscopy Sciences). Samples were permeabilized in 0.1% Triton-X-100 for 3 min and then incubated in a blocking solution consisting of 10% FBS (Sigma-Aldrich), 0.01% sodium azide (Sigma-

Aldrich), 1× PBS for 1 h at room temperature (RT). After three washes in 1× PBS, the cells were stained with Alexa 647-conjugated antibody in blocking solution (20 μg/ml) for 1 h at RT. The cells were washed 3× in PBS and imaged using TIRF-SIM or TIRF microscope as detailed above.

**Image analysis**. Image J was used to measure the distribution of molecules at microclusters in multi-color TIRF-SIM images as shown in Fig. 1 and Supplementary Fig. 1 by drawing a line scan across a microcluster, and then measuring the line scan intensity for each channel. After background subtraction, the line scans were normalized for intensity. For kinetic analysis of microclusters, the Time Series Analyzer plugin in Image J was used. A boxed region of interest was cropped over a microcluster and fluorescence intensity over time for all colors was obtained and normalized for intensity. Plotting the normalized intensity of each channel over time resulted in microcluster recruitment graphs as shown in Fig. 2. To measure the kinetic lag between two molecules, the normalized intensity data was exported to the Prism7 software (Graphpad) and analyzed using sigmoidal non-linear fit. The resulting $R^2$ was monitored for proper fitting of normalized intensity data. The Kinetic lag was obtained by calculating the difference in the half-max of non-linear fitted curves. Kymographs in Fig. 5b, i were created using Image J.

## Data availability

Data supporting the findings of this study are available within the article and its Supplementary Information files or from the corresponding author upon reasonable request.

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

## Acknowledgements

This research was supported by the Intramural Research Program of the NIH, NCI, CCR. We thank the Advanced Imaging Center (AIC) at Janelia Research Campus for access to the TIRF-SIM microscope. The AIC is jointly supported by the Howard Hughes Medical Institute and the Gordon and Betty Moore Foundation. We thank Xufeng Wu and NHLBI for access to the Delta Vision OMX microscope. We thank Luke Lavis at Janelia research campus for his generous contribution of Halo-tag ligands. We thank Paul Randazzo for input on kinetic analysis. We thank Art Weiss and Wan-Lin Lo for sharing J.LAT KO cells.

## Author contributions

J.Y., L.B., T.N., and K.M.M. performed the experiments; J.Y. and L.E.S designed the study; J.Y. and L.B. performed image analysis; J.Y. prepared figures and wrote the manuscript with help from L.B. and comments from L.E.S.

## Additional information

**Competing interests:** The authors declare no competing interests.

