## [Peer Review File · Nature Communications]

Reviewers' comments:

Reviewer #1 (Immunological synapse, TCR signaling)(Remarks to the Author):

J. Yi et al. used TIRF-SIM to investigate the dynamics of TCR microclusters and associated signaling components in Jurkat T cells stimulated by immobilized anti-CD3 ϵ antibodies. They report the formation of two spatially distinct domains: the TCR/CD3/ZAP-70 complex and the LAT-associated signaling complex. Kinetic analysis of microcluster formation point out a stepwise recruitment of ZAP-70 and downstream signaling components to the TCR. The analysis also shows that delays in stepwise recruitment of microcluster components are dependent on $[Ca^{2+}]_i$ increase.

The manuscript is interesting and is based on cutting-edge imaging techniques. It addresses an important unresolved question in T lymphocyte biology concerning the kinetics of TCR-mediated signal components assembly following productive TCR engagement. The results reported in this study are thought-provoking and can be instrumental to revise current models of T lymphocyte activation. However, the study has, to my opinion, some weak points.

1) One concern I have is about the variability in the time-kinetics of recruitment of each signaling component as shown in the different plots. For instance, in Figure 2, ZAP-70 relative intensity reaches a plateau within ~50 seconds for panel E; within ~80 seconds for panel B; within ~150 second for panel D. Similar variability is observed when comparing other plots showing results for other signaling components. What is the reason of such variability? Can this variability jeopardize conclusions?

2) Results shown in Movie 3 are not fully convincing. SLP76 appears to be already present in the TIRF plane since the beginning of the movie. I was not able to detect a delay in the recruitment of this signaling component when compared to ZAP-70.

3) Results on the role of $[Ca^{2+}]_i$ increase in delaying microcluster recruitment are puzzling. $[Ca^{2+}]_i$ increase or calcium chelation might be altered at room temperature. It is important to confirm results with experiments performed at 37°C. It would be also interesting to evaluate the impact that treatment of Jurkat cells with ionomycin might have on microcluster recruitment. This experiment would allow to define whether a $[Ca^{2+}]_i$ increase preceding TCR signaling might affect the upstream signaling cascade.

4) The study is somehow limited in scope. Jurkat cells are an useful cellular tool to study TCR coupled signal transduction and have the advantage to be relatively easy to be transfected and to be inspected using imaging systems. Yet, it is not clear whether the reported findings might reflect the activation of untransformed human or mouse T cells. It would be important that at least part of the reported data would be reproduced in a more physiologically relevant cellular system. For instance the role of Ca^{2+} chelators on ZAP-70 microcluster delay might be investigated in non-transformed cells.

5) Movie legends are missing; they should be included in the manuscript.

Reviewer #2 (Thymic selection, TCR signaling)(Remarks to the Author):

This paper uses structured illumination total internal reflection fluorescence microscopy (TIRF-SIM), which allows very high resolution (sub-diffraction), time-lapse imaging, to analyze molecular movements at the immunological synapse during T cell activation. The overall finding is that the recruitment of different parts of the signaling machinery occurs at different times, and that there are

clear demarcations between the receptor part of the synapse (defined by CD3z-bound Zap70) and the LAT signalosome (signaling) part of the synapse. There is a clear time-lag in recruitment between proteins associated with these two different types of microcluster.

The experimental system is based on Jurkat cells activated by plate-bound anti-TCR antibodies (which would have precluded its funding by NIH outside of the NIH intramural system, but I digress). Using Jurkat and α -TCR is certainly a limitation. The first interesting result is that the recognition and signaling domains are separate. This is very clear and is discussed in terms of the various models and experimental data that are already in the literature. This is all fine. The result is reminiscent of the findings from Griffiths' lab about the separation of the recognition part of the synapse and the effector part where the cytolytic granules are released where a CTL interfaces with its target. This should be mentioned.

The recognition molecules are recruited before the signaling components, but the signaling components are all recruited at the same time. This was a little surprising, but the data are clear. They are explained in terms of the "highly cooperative nature of protein-protein interactions". They note that cCbl is recruited relatively later, and that its peak coincides with a reduction of Grb2. This makes sense, but the data referred to (Fig 4a) do not seem to show this, at least not as far as I can see. The arrow on the fig points to the peak of cCbl recruitment, but for Grb2, it looks to me that it is also at the peak.

There is no lag between CD3 and ZAP in the earliest microclusters, but a lag is noticeable in later-formed microclusters. Could it be that zeta is already bound to Zap70 in those early clusters? Analysis of Ca^{2+} flux showed that the change to longer lag-times happened at the point where Ca^{2+} flux occurred, and they then showed that the lag seemed indeed to be regulated by cytoplasmic Ca^{2+} .

The experiments are well performed and seem to have all the appropriate controls. The study raises a lot of questions for the future, without really resolving any, but the findings are certainly of interest.

Reviewer #3 (TCR signaling, costimulation)(Remarks to the Author):

The paper by Yi and co-authors analyzed T cell receptor (CTCR) micro clusters, which are calcium-dependent due to T cell activation. Using internal reflection fluorescence fractured illumination microscopy (TIRF-SIM) the authors determined the organization and composition of TCR micro clusters thereby identifying spatial distinct domains, which have a distinct kinetic due to calcium influx mediated upon T cell activation. Therefore, this manuscript extends the current knowledge of the evolution of TCR micro clusters during T cell activation. Despite few novel insights into TCR micro clusters, there exist some queries throughout the manuscript:

- The paper extends the information published by the same group (Balagopalan et al., 2018). Thus, the novelty of the data shown in comparison of their recent publication should be stated more obvious.
- Jurkat T cells were used as model for TCR micro clusters analyses, data on primary T cells are required to underpin the importance of this finding.
- A calcium influx inhibitor should be used as a control.
- To receive more data on the functional impact the use of immune checkpoint inhibitors or negative signaling on T cell activation experiments and recruitment of signaling clusters is required.
- Calcium influx should have been determined overtime.
- Analysis of ZAP70 phosphorylation during calcium influx signaling is required.
- Citation Balagopalan et al., 2018 is wrong, please provide further information.

Reviewers' comments:

Reviewer #1 (Immunological synapse, TCR signaling)(Remarks to the Author):

J. Yi et al. used TIRF-SIM to investigate the dynamics of TCR microclusters and associated signaling components in Jurkat T cells stimulated by immobilized anti-CD3 ϵ antibodies. They report the formation of two spatially distinct domains: the TCR/CD3/ZAP-70 complex and the LAT-associated signaling complex. Kinetic analysis of microcluster formation point out a stepwise recruitment of ZAP-70 and downstream signaling components to the TCR. The analysis also shows that delays in stepwise recruitment of microcluster components are dependent on [Ca²⁺]_i increase.

The manuscript is interesting and is based on cutting-edge imaging techniques. It addresses an important unresolved question in T lymphocyte biology concerning the kinetics of TCR-mediated signal components assembly following productive TCR engagement. The results reported in this study are though-provoking and can be instrumental to revise current models of T lymphocyte activation.

However, the study has, to my opinion, some weak points.

1) One concern I have is about the variability in the time-kinetics of recruitment of each signaling component as shown in the different plots. For instance, in Figure 2, ZAP-70 relative intensity reaches a plateau within ~50 seconds for panel E; within ~80 seconds for panel B; within ~150 second for panel D. Similar variability is observed when comparing other plots showing results for other signaling components. What is the reason of such variability? Can this variability jeopardize conclusions?

The specific variability among the kinetic curves that the reviewer points out in Figure 2 is mostly due to different starting points for each microcluster, which in turn is a function of when the cell makes contact with the coverslip. As we are analyzing the formation of individual microclusters at the synapse, there will be varied starting points for each microcluster.

However, we would like to note that, there is also variability in microcluster formation rates measured by increase in the corresponding fluorescent signal (i.e. Hill slope of the fitted sigmoidal curve). The rate of signal increase could be a useful parameter in future analysis and could inform rates of association, but does not affect our kinetic lag analysis. As stated in the results and methods sections, the kinetic lag is a measure of the delta between the half max of each of the two fitted sigmoidal curves regardless of the slope of each curve and is not affected by the Hill slope of each curve. Importantly, the experimental data showed the kinetic lags between zap and zeta or Grb2 and Zap70 were remarkably consistent at ~27s at 21°C across many cells. Moreover, similar kinetic lags were shown to be present in primary human T cells and at physiological temperature.

2) Results shown in Movie 3 are not fully convincing. SLP76 appears to be already present in the TIRF plane since the beginning of the movie. I was not able to detect a delay in the recruitment of this signaling component when compared to ZAP-70.

Thank you for the comment. It informed us that more detailed description of movie 3 was required for the general audience. As the reviewer noted, there are already older SLP76 clusters in the central part of the cell, but at the periphery, ZAP70 clusters are starting to form and have formed without neighboring SLP76. Through the course of the movie, SLP76 clusters began to form near the preformed ZAP70 clusters. We have updated this movie by inserting a zoomed-in region, where the delay between ZAP70 and SLP76 clusters are clearly visible.

3) Results on the role of $[Ca^{2+}]_i$ increase in delaying microcluster recruitment are puzzling. $[Ca^{2+}]_i$ increase or calcium chelation might be altered at room temperature. It is important to confirm results with experiments performed at 37°C. It would be also interesting to evaluate the impact that treatment of Jurkat cells with ionomycin might have on microcluster recruitment. This experiment would allow to define whether a $[Ca^{2+}]_i$ increase preceding TCR signaling might affect the upstream signaling cascade.

As per the reviewer's request, we have now confirmed the effects of calcium chelation using BAPTA and EGTA at 37°C. We observed a significant decrease in kinetic lags between TCR ζ -ZAP70 and ZAP70-GRB2 in late-formed microclusters recapitulating what was observed at 21°C. These results are included in Supplementary Figure 4A of the revised manuscript. Also as the reviewer requested, we have also evaluated the impact of the calcium ionophore ionomycin on microcluster recruitment. Converse to the effects of calcium chelation, calcium elevation via ionomycin led to a significant increase in kinetic lags between TCR ζ -ZAP70 and ZAP70-GRB2. These results are included in Fig. 6D of the revised manuscript. We have also included calcium flux analysis by flow over time to corroborate the imaging data from various calcium concentration conditions shown in Figure 6 C and D. The data confirms that the calcium chelators (BAPTA/EGTA and GCamp6) and calcium ionophore conversely affect calcium influx in the cells as predicted (Supp. Fig 4B). We believe that these results strengthen our observation that the stepwise recruitment of microcluster components is dependent on calcium.

4) The study is somehow limited in scope. Jurkat cells are an useful cellular tool to study TCR coupled signal transduction and have the advantage to be relatively easy to be transfected and to be inspected using imaging systems. Yet, it is not clear whether the reported findings might reflect the activation of untransformed human or mouse T cells. It would be important that at least part of the reported data would be reproduced in a more physiologically relevant cellular system. For instance the role of Ca^{2+} chelators on ZAP-70 microcluster delay might be investigated in non-transformed cells.

As per this reviewer's request we have now verified the stepwise recruitment of microcluster components in nontransformed human cells. CD3⁺ lymphoblasts from human peripheral blood were transfected with TCR ζ -Emerald and ZAP70-Scarlet or ZAP70- Emerald and Grb2-Scarlet and imaged on coverslips coated with anti-CD3 and anti-CD28 at 37°C. As observed in Jurkat cells, primary human T cell blasts exhibited delays in the TIRF time-lapse images between the recruitment of ZAP70-Scarlet to TCR ζ -Emerald and Grb2-Emerald to ZAP70-scarlet. A kinetic lag of ~17s in ZAP70 recruitment to TCR ζ and ~12s in GRB2 to ZAP70 recruitment was observed. Thus, although the kinetics are slightly different from Jurkat cells, the stepwise recruitment of microcluster proteins was confirmed in primary human lymphoblasts at

physiological temperature. These results are now included in Figure 2H-J of the revised manuscript.

5) Movie legends are missing; they should be included in the manuscript.

Noted. Have been added.

Reviewer #2 (Thymic selection, TCR signaling)(Remarks to the Author):

This paper uses structured illumination total internal reflection fluorescence microscopy (TIRF-SIM), which allows very high resolution (sub-diffraction), time-lapse imaging, to analyze molecular movements at the immunological synapse during T cell activation. The overall finding is that the recruitment of different parts of the signaling machinery occurs at different times, and that there are clear demarcations between the receptor part of the synapse (defined by CD3z-bound Zap70) and the LAT signalosome (signaling) part of the synapse. There is a clear time-lag in recruitment between proteins associated with these two different types of microcluster.

The experimental system is based on Jurkat cells activated by plate-bound anti-TCR antibodies (which would have precluded its funding by NIH outside of the NIH intramural system, but I digress). Using Jurkat and a-TCR is certainly a limitation. The first interesting result is that the recognition and signaling domains are separate. This is very clear and is discussed in terms of the various models and experimental data that are already in the literature. This is all fine. The result is reminiscent of the findings from Griffiths' lab about the separation of the recognition part of the synapse and the effector part where the cytolytic granules are released where a CTL interfaces with its target. This should be mentioned.

This is a good suggestion. Citation has been added to the appropriate introduction and discussion sections (also see comments in reviewer 3). However, we would like to note that the effector response in the Griffiths lab paper refers to a much later process in T cell activation, which involves a complete reorganization of the cell's cytoskeleton, organelles, and exocytic machinery which occur in the order of minutes, and is understood to be much further downstream than the proximal TCR signaling pathways discussed in our paper. The surprise for us was the kinetic lags between proximal signaling components which have been thought to cluster more or less simultaneously upon TCR ligation.

The recognition molecules are recruited before the signaling components, but the signaling components are all recruited at the same time. This was a little surprising, but the data are clear. They are explained in terms of the "highly cooperative nature of protein-protein interactions". They note that cCbl is recruited relatively later, and that its peak coincides with a reduction of Grb2. This makes sense, but the data referred to (Fig 4a) do not seem to show this, at least not as far as I can see. The arrow on the fig points to the peak of cCbl recruitment, but for Grb2, it looks to me that it is also at the peak.

Thank you for the comment. This informed us that the text was not clear in describing the relationship between GRB2 (and other signaling molecules) and c-Cbl. First, we meant to convey that there is a short lag in time between recruitment of signaling domain components

(represented by Grb2 in Fig 4a) and c-Cbl, which is measured by the delta in the half-max of each fitted curve. This short kinetic lag is visually apparent by the offset between the red (GRB2) and green (c-Cbl) curves during the recruitment phase of the graph (until ~100s).

The second observation has to do with the right side of the graph which shows rapid decay in signals for both GRB2 and c-Cbl. The beginning of the rapid decay coincides with the peak in recruitment of c-Cbl which is marked by the black arrowhead. GRB2 is also near its peak, but rapidly begins to lose signal, indicating cluster disassociation. As the reviewer pointed out, GRB2 and c-Cbl signals were not inversely correlated. We have revised the text in Results section and figure legend to clarify the relationship between GRB2 and c-Cbl, to indicate that they are not inversely correlated, rather that the peak in c-Cbl signal corresponds to the beginning of the rapid decay in signals for both.

There is no lag between CD3 and ZAP in the earliest microclusters, but a lag is noticeable in later-formed microclusters. Could it be that zeta is already bound to Zap70 in those early clusters? Analysis of Ca²⁺ flux showed that the change to longer lag-times happened at the point where Ca²⁺ flux occurred, and they then showed that the lag seemed indeed to be regulated by cytoplasmic Ca²⁺.

This is an intriguing idea and is worth pursuing in future experiments. Moreover, it is consistent with previous studies showing that TCR exist as pre-formed clusters in un-activated T cells (*Crites et al., 2014 J Immunol 193:56-67; Thill et al., 2016 Mol Cell Biol 36:2396-2402; van Oers et al., 1994 Immunity 1:675-685*).

The experiments are well performed and seem to have all the appropriate controls. The study raises a lot of questions for the future, without really resolving any, but the findings are certainly of interest.

Reviewer #3 (TCR signaling, costimulation)(Remarks to the Author):

The paper by Yi and co-authors analyzed T cell receptor (CTCR) micro clusters, which are calcium-dependent due to T cell activation. Using internal reflection fluorescence fractured illumination microscopy (TIRF-SIM) the authors determined the organization and composition of TCR micro clusters thereby identifying spatial distinct domains, which have a distinct kinetic due to calcium influx mediated upon T cell activation. Therefore, this manuscript extends the current knowledge of the evolution of TCR micro clusters during T cell activation. Despite few novel insights into TCR micro clusters, there exist some queries throughout the manuscript:
- The paper extends the information published by the same group (Balagopalan et al., 2018). Thus, the novelty of the data shown in comparison of their recent publication should be stated more obvious.

As suggested by reviewers 2 and 3, we have added the Balagopalan et al 2018 and the Griffiths lab paper citations to the Introduction and Discussion sections and have indicated that those studies have focused, not on early events, but on the kinetics of late stage T cell activation

processes such as vesicular recruitment and effector function. In the current study we are looking at the proximal, early TCR signaling events, which have not been characterized by those studies.

- Jurkat T cells were used as model for TCR micro clusters analyses, data on primary T cells are required to underpin the importance of this finding.

As per this reviewer's request we have now verified the stepwise recruitment of microcluster components in nontransformed human cells. CD3⁺ lymphoblasts from human peripheral blood were transfected with TCR ζ -Emerald and ZAP70-Scarlet or ZAP70- Emerald and Grb2-Scarlet and imaged on coverslips coated with anti-CD3 and anti-CD28 at 37°C. As observed in Jurkat cells, primary human T cell blasts exhibited delays in the TIRF time-lapse images between the recruitment of ZAP70-scarlet to TCR ζ -Emerald and Grb2-Emerald to ZAP70-Scarlet. A kinetic lag of ~17s in ZAP70 recruitment to TCR ζ and ~12s in GRB2 to ZAP70 recruitment was observed. Thus, although the kinetics are slightly different from Jurkat cells, the stepwise recruitment of microcluster proteins was confirmed in primary human lymphoblasts at physiological temperature. These results are now included in Figure 2H-J of the revised manuscript.

- A calcium influx inhibitor should be used as a control.

We point the reviewer to Fig. 6C and D in which GCam6 and BAPTA-EGTA were used as calcium influx inhibitors. Both treatments caused a decrease in kinetic lags between TCR ζ -Emerald and ZAP70-Scarlet or ZAP70- Emerald and Grb2-Scarlet. We have also confirmed that there is no calcium flux in BAPTA/EGTA treated cells, and that cells expressing GCamp6 show decreased calcium flux when in solution with 2mM CaCl₂ (used in all the imaging experiments), and that calcium flux is rescued by addition of 5mM CaCl₂ to GCamp6 expressing cells (Supp. Fig 4B).

- To receive more data on the functional impact the use of immune checkpoint inhibitors or negative signaling on T cell activation experiments and recruitment of signaling clusters is required.

We have generated data, which we believe provide a significant increase in our understanding of the structure and kinetics of formation of microclusters. Our kinetic results have multiple implications for models of TCR signal transduction. We agree with the reviewer that looking at the effects of immune checkpoint inhibitors and negative signaling on signaling cluster recruitment is an important next step that we certainly plan to pursue. We also plan to look at the effects of costimulation, agonist strength and ubiquitinylation on stepwise recruitment and disassociation of microclusters. In particular, a mechanistic study of the role of co-stimulatory and inhibitory signaling domains on microcluster recruitment kinetics will require knockdown/mutation analysis of receptors (PD-1, CTLA-4, PD-1, CD28, ICOS), phosphatases (e.g. SHP1, SHP2), kinases, as well as generation of new reporters to evaluate the effects of these modulations. Therefore, we strongly believe that experiments to modulate kinetic recruitment parameters by stimulatory or inhibitory conditions is beyond the scope of this paper.

- Calcium influx should have been determined overtime.

We point the reviewer to Fluo4 imaging shown in Fig. 5A and movie 4. We have also included in the revised version of the manuscript, calcium flux analysis by flow over time to corroborate the imaging data from various calcium concentration conditions shown in Figure 6 C and D. We have confirmed that there is no calcium flux in BAPTA/EGTA treated cells, elevated levels of calcium in ionomycin treated cells and that cells expressing GCamp6 show decreased calcium flux when in solution with 2mM CaCl₂ (used in all the imaging experiments), and that calcium flux is rescued by addition of 5mM CaCl₂ to GCamp6 expressing cells (Supp. Fig 4B).

- Analysis of ZAP70 phosphorylation during calcium influx signaling is required.

We have data looking at ZAP70 phosphorylation by fixed cell imaging in DMSO and BAPTA/EGTA treated cells. These data show that phosphorylation of endogenous ZAP70 on Y493 is elevated at early time points (2min) indicating a faster kinetic of ZAP70 activation. This is consistent with the decrease in kinetic lag of ZAP70 recruitment to TCR ζ in BAPTA/EGTA treated cells. This data is currently included in a manuscript in preparation by our collaborator. We can certainly include these data if required in a confidential letter to the reviewer, but trust that s/he understands we cannot include these results in the revised manuscript.

- Citation Balagopalan et al., 2018 is wrong, please provide further information.

Citation has been corrected.

REVIEWERS' COMMENTS:

Reviewer #1 (Remarks to the Author):

The authors adequately addressed the points I raised. In particular the new data obtained using human non-transformed cells, performing measurements at 37°C and adding ionomycin to cells are interesting. The manuscript is substantially improved.

There is one minor point to address. In Figure S4 it is indicated that the light green curve corresponds to 2mM CaCl₂ while in the legend it is written that the light green curve corresponds to 1mM CaCl₂. This must be a typo.

This incongruence also appears elsewhere. For instance in reply to points raised by Reviewer 3 the authors state that 2mM CaCl₂ was used in all the imaging experiments (obviously except titration experiments). Conversely, in Materials and Methods section it is stated that experiments were performed with 1mM CaCl₂.

It would be important to clarify this point.

Reviewer #2 (Remarks to the Author):

The authors have revised the manuscript and provided new data on primary T cells. They have responded to all of my criticisms, and I believe, to those of the other reviewers. This is an interesting study.

Reviewer #3 (Remarks to the Author):

Yi and co-authors analyzed the assembly of micro clusters of T cell receptors and its regulation by calcium. The authors demonstrate a spatial and kinetic regulation of the TCR micro clusters during activation, which is controlled by calcium flux. The authors comprehensively worked on the manuscript and included novel data regarding the recruitment of micro cluster components. In addition, they included analyses of human primary T cells. They modulated calcium concentrations and performed the time kinetics requested. The primary T cell data are described and mainly confirm the results of Jurkat cells despite a distinct kinetic formation of micro clusters. However, it is pity that the effect of checkpoint inhibitors has not been analyzed in this context, which would have even more improved the manuscript.

REVIEWERS' COMMENTS:

Reviewer #1 (Remarks to the Author):

The authors adequately addressed the points I raised. In particular the new data obtained using human non-transformed cells, performing measurements at 37°C and adding ionomycin to cells are interesting. The manuscript is substantially improved.

There is one minor point to address. In Figure S4 it is indicated that the light green curve corresponds to 2mM CaCl₂ while in the legend it is written that the light green curve corresponds to 1mM CaCl₂. This must be a typo.

This incongruence also appears elsewhere. For instance in reply to points raised by Reviewer 3 the authors state that 2mM CaCl₂ was used in all the imaging experiments (obviously except titration experiments). Conversely, in Materials and Methods section it is stated that experiments were performed with 1mM CaCl₂.

It would be important to clarify this point.

We thank the Reviewer for supporting the publication of our manuscript and also for pointing out the inconsistency in labeling of CaCl₂ concentrations. We have now corrected the figure legend for Figure S4 as well as the Methods section to both read 2mM CaCl₂.

Reviewer #2 (Remarks to the Author):

The authors have revised the manuscript and provided new data on primary T cells. They have responded to all of my criticisms, and I believe, to those of the other reviewers. This is an interesting study.

We thank the Reviewer for supporting the publication of our manuscript.

Reviewer #3 (Remarks to the Author):

Yi and co-authors analyzed the assembly of micro clusters of T cell receptors and its regulation by calcium. The authors demonstrate a spatial and kinetic regulation of the TCR micro clusters during activation, which is controlled by calcium flux. The authors comprehensively worked on the manuscript and included novel data regarding the recruitment of micro cluster components. In addition, they included analyses of human primary T cells. They modulated calcium concentrations and performed the time kinetics requested. The primary T cell data are described and mainly confirm the results of Jurkat cells despite a distinct kinetic formation of micro clusters. However, it is pity that the effect of checkpoint inhibitors has not been analyzed in this context, which would have even more improved the manuscript.

We thank the Reviewer for supporting the publication of our manuscript.